# Precision Medicine in Head and Neck Cancers: Genomic and Preclinical Approaches

**DOI:** 10.3390/jpm12060854

**Published:** 2022-05-24

**Authors:** Giacomo Miserocchi, Chiara Spadazzi, Sebastiano Calpona, Francesco De Rosa, Alice Usai, Alessandro De Vita, Chiara Liverani, Claudia Cocchi, Silvia Vanni, Chiara Calabrese, Massimo Bassi, Giovanni De Luca, Giuseppe Meccariello, Toni Ibrahim, Marco Schiavone, Laura Mercatali

**Affiliations:** 1Osteoncology Unit, Bioscience Laboratory, IRCCS Istituto Romagnolo per lo Studio dei Tumori (IRST) “Dino Amadori”, 47014 Meldola, Italy; giacomo.miserocchi@irst.emr.it (G.M.); alessandro.devita@irst.emr.it (A.D.V.); chiara.liverani@irst.emr.it (C.L.); claudia.cocchi@irst.emr.it (C.C.); silvia.vanni@irst.emr.it (S.V.); chiara.calabrese@irst.emr.it (C.C.); laura.mercatali@irst.emr.it (L.M.); 2Clinical and Experimental Oncology, Immunotherapy, Rare Cancers and Biological Resource Center, IRCCS Istituto Romagnolo per lo Studio dei Tumori (IRST) “Dino Amadori”, 47014 Meldola, Italy; sebastiano.calpona@irst.emr.it (S.C.); francesco.derosa@irst.emr.it (F.D.R.); 3Department of Biology, University of Pisa, 55126 Pisa, Italy; a.usai@studenti.unipi.it; 4Maxillofacial Surgery Unit, “Bufalini Hospital”, AUSL Romagna, 47521 Cesena, Italy; massimo.bassi@auslromagna.it; 5Pathology Unit, “Bufalini” Hospital, AUSL Romagna, 47521 Cesena, Italy; giovanni.deluca@auslromagna.it; 6Otolaryngology and Head-Neck Surgery Unit, Department of Head-Neck Surgeries, Morgagni Pierantoni Hospital, AUSL Romagna, 47121 Forlì, Italy; giuseppe.meccariello2@auslromagna.it; 7Osteoncology, Bone and Soft Tissue Sarcomas and Innovative Therapies Unit, IRCCS Istituto Ortopedico Rizzoli, 40136 Bologna, Italy; toni.ibrahim@ior.it; 8Department of Molecular and Translational Medicine, University of Brescia, 25123 Brescia, Italy; marco.schiavone@unibs.it

**Keywords:** head and neck cancer, multi-omic analysis, 3D culture, patient-derived xenograft, zebrafish

## Abstract

Head and neck cancers (HNCs) represent the sixth most widespread malignancy worldwide. Surgery, radiotherapy, chemotherapeutic and immunotherapeutic drugs represent the main clinical approaches for HNC patients. Moreover, HNCs are characterised by an elevated mutational load; however, specific genetic mutations or biomarkers have not yet been found. In this scenario, personalised medicine is showing its efficacy. To study the reliability and the effects of personalised treatments, preclinical research can take advantage of next-generation sequencing and innovative technologies that have been developed to obtain genomic and multi-omic profiles to drive personalised treatments. The crosstalk between malignant and healthy components, as well as interactions with extracellular matrices, are important features which are responsible for treatment failure. Preclinical research has constantly implemented in vitro and in vivo models to mimic the natural tumour microenvironment. Among them, 3D systems have been developed to reproduce the tumour mass architecture, such as biomimetic scaffolds and organoids. In addition, in vivo models have been changed over the last decades to overcome problems such as animal management complexity and time-consuming experiments. In this review, we will explore the new approaches aimed to improve preclinical tools to study and apply precision medicine as a therapeutic option for patients affected by HNCs.

## 1. Introduction

Head and neck cancer (HNC) is the sixth most common non-skin cancer worldwide, resulting in around 900,000 new cases each year, with around 460,000 new deaths (50% mortality rate) [1,2]. Squamous cell carcinomas (SCCs) constitute around 90% of total HNCs, and represent a subset of malignancies which arise in the squamous epithelial cells located in the mucosa of the oropharynx, hypopharynx, nasopharyngeal, oral and nasal cavity, larynx, and salivary glands [2,3]. Alcohol, tobacco, genetic mutations, viral infections of human papilloma virus (HPV) and Epstein-Barr virus (EBV) represent the major risk factors of HNCs [2,3,4,5]. HPV-positive SCCs mainly arise in the oropharyngeal tissues, and show better clinical outcomes and different pathological characteristics compared to their HPV-negative counterparts [6,7]. In contrast, EBV-positive SCCs are associated with nasopharyngeal tumours, especially in east and southeast Asia, where this type of carcinoma is endemic [8,9]. 

HNSCC (head and neck squamous cell carcinoma) carcinogenesis includes a series of pathological steps involving different kinds of cell types, according to the origin site of the neoplasia. Usually, the disease arises with the development of epithelial cells hyperplasia, which results in mild, moderate, or severe dysplasia [8]. The progression to neoplasia can be driven by different cell types, given the heterogeneous nature of HNSCCs. Carcinogenesis is driven by gene mutations and chromosomal instability, which could play a different role in the case of a presence/absence of HPV or EBV infection [10]. Thus, mutations in *TP53*, *PIK3CA*, *FAT1,* and *NOTCH*, the amplification of *CCND1,* and a loss of *CDKN2A* represent the most common genetic alterations in HPV-negative HNSCCs [10,11]. On the other hand, HPV-positive HNSCCs show different primary genetic alterations, including the amplification of *E2F1*, a loss of *TRAF3,* and a higher rate of *PIK3CA* mutations [10,11]. Tobacco and alcohol abuse represent the two principal risk factors for the onset of HPV-negative HNSCCs. Indeed, most of the chemicals present in tobacco have been demonstrated to have carcinogenic properties. Metabolic activation and inflammation induced by tobacco-related compounds are two events involved in the dysregulation of DNA repair processes, and are responsible for oncogenic transformation leading to genetic mutations and abnormalities [12]. Instead, alcohol is metabolized to acetaldehyde, a substance which is able to develop DNA adducts, as well as to synergize with tobacco to promote carcinogenesis [13,14]. Conversely, HPV-related HNSCCs show different pathological features in terms of gene expression, immune profiles, and genetic alterations. The 70% of HPV-positive HNSCCs arise in oropharynx tissues, showing lower percentages in other sites [15]. HPV-16 represents the most widespread virus strain implicated in HNCs. Among the nine proteins produced by the viral genome, E6 and E7 can induce malignant transformation [7,16]. E6 forms complexes with the tumour suppressor p53, promoting its ubiquitylation and subsequent proteasomal degradation [8,16]. The alteration of p53 activity is observed also in HPV-negative HNCs, but it has different causes, since it depends on mutations or deletions of the *TP53* gene [17,18]. Moreover, E7 promotes the proteasomal degradation of retinoblastoma-associated protein (Rb1), a cell cycle regulator [18]. The degradation of Rb1 involves the upregulation of p16, an inhibitor of cyclin-dependent kinases (CDK), which is a well-known marker of HPV+ tumours [18,19].

However, other elements are involved in tumour progression. In this scenario, carcinogenesis is mainly induced by the oncogenic transformation of either adult stem cells or progenitor cells into cancer stem cells (CSCs) [20]. CSCs are a minority subpopulation of all tumour subclones distinguished by self-renewal properties and the capability to generate differentiated progeny, characterised by only transient proliferative status [21]. The cell adhesion molecule CD44 and aldehyde dehydrogenase (ALDH), an enzyme responsible for the oxidisation of aldehydes to carbolic acid, represent two markers of CTCs phenotypes in numerous solid tumours [21,22,23]. Depending on the HPV status, HNSCCs show differences in the plasticity and frequency of CTCs [24,25]. Several studies report on CSCs frequency in HNSCC tissues, but the results are still contradictory, describing different levels of stem cells in both HPV statuses [25,26]. The self-renewal properties, typical of these kinds of cells, are one of the causes of their resistance to standard therapies [27]. The characterization of HNSCC CSCs has attempted to identify biomarkers of chemo- and radioresistance. Chromatin regulator Bmi-1 and CD133 are only two examples of drug-sensitivity markers of CSCs, but their expression does not discriminate between HPV-positive and negative patients [28,29,30]. The higher efficacy of CSCs to repair the DNA damage is the main cause of radioresistance. Biomarkers such as RAD51, involved in the homologous recombination and repair of DNA, are highly expressed in HNSCC CSCs [22,31]. A promising marker involved in the higher radiotherapy efficacy typical of HPV-positive HNSCCs is represented by the fluorescent fusion protein ZsGreen-cODC. This protein, indicative of tumorigenic cells, is associated with mechanisms of radiation-induced dedifferentiation in HNSCC cell lines. Indeed, the ZsGreen-cODC-negative cells of HPV-negative cell lines showed higher re-expression of ZsGreen-cODC and higher dedifferentiation after radiation, all of which are features related to radioresistance [26].

Cell surface signalling receptors are other important biomarkers frequently used as targets for tumour treatment [32]. Among them, epithelial growth factor receptor (EGFR), an ErbB tyrosine kinase receptor family member, represents an important target for HNC treatment [33]. SCCs cells show a hyperactivation of EGFR signalling, a genetic event involved in the deregulation of several processes, including proliferation, angiogenesis, and metastasis development [8,34]. The EGFR expression is significantly downregulated with increasing distance from tumour microvessels [35]. Indeed, hypoxia has been identified as one of the major causes of treatment resistance of different tumours, including HNCs [35,36]. The hypoxic microenvironment induces changes in the DNA and growth properties of cancer cells, which are features associated with chemo- and radioresistance [35,36,37,38]. Oxygen represents an ideal radiosensitizer, due to its ability to induce the fixation of radical-induced DNA damage [38,39]. One of the major modulators of the adaptation response of tumour cells to low oxygen levels is hypoxia-inducible factor (HIF-1) [40]. HIF-1 activation promotes the transcription of over 100 downstream genes involved in tumour progression and survival [41]. For all of these reasons, HIF-1 and its related genes may be promising biomarkers of HNCs, as well as therapeutic targets for innovative therapies.

## 2. HNCs Clinical Approaches 

HNCs are a heterogeneous group of malignancies that make a personalised therapeutic approach necessary for each patient. As a matter of fact, clinicians have to evaluate several parameters before therapy selection, such as tumour site, stage, natural history of the disease, biological characteristics, and clinical status. The curative approach is highly effective in preserving the organ’s function. In locoregional or local disease, therapeutic options include surgery, radiotherapy, and systemic chemotherapy [6,9]. Oral cavity cancers are commonly resected, while sites such as the larynx and pharynx are treated with standard or hypo-fractionated radiation [42]. To better preserve the tissue integrity with a minimally invasive approach, robotic or laser resection and the involvement of reconstructive techniques represent advanced surgical treatments, especially in larynx tumours [43]. Malignancies at a high risk of recurrence are commonly treated with postoperative radiation or chemoradiotherapy (CRT) in order to prevent relapses and to increase patient survival [9,44]. The involvement of surgical margins, extra-nodal extension, and perineural invasion are pathological features that describe the group of patients with a higher risk of recurrence. In these cases, radiotherapy and high doses of cisplatin-based chemotherapy improve patient survival [45]. Conversely, tumours at an advanced stage are commonly treated with CRT. The chemotherapeutic approach is based on high dose cisplatin-based monotherapy (100 mg/m^2^) or on low dose treatment in patients affected by renal dysfunctions or hearing loss [46]. Cetuximab, an epithelial growth factor receptor (EGFR) monoclonal antibody (mAbs), has been demonstrated to improve the effectiveness of radiation treatment [46,47]. The restricted group of EBV-positive nasopharyngeal cancers display not only treatment sensitivity, but also a higher risk of metastasis development [9]. An improvement in survival was demonstrated in EBV-positive patients treated with induction therapy or with additional chemotherapeutic regimens after CRT [48]. The combination of different drugs represents one of the front-line options of treatment in recurrent or metastatic HNSCC settings [49,50]. In this scenario, clinical and preclinical research needs to identify new treatment approaches. Recently, immunotherapy (IT) has been studied as an alternative option for the treatment of HNSCCs. Immunotherapeutic agents act directly on the patients’ immune system-activated cells to fight the disease. Programmed death 1 (PD1) immune check-point inhibitors have shown significant effectiveness compared to standard treatments [50,51]. In 2016, the Food and Drug Administration (FDA) approved two anti-PD-1 antibodies, nivolumab and pembrolizumab, for the treatment of recurrent or metastatic HNCs based on the results of the phase I KEYNOTE-12, phase II KEYNOTE-055, phase III KEYNOTE-040, and phase III CheckMate 141 trials [2,50,51]. Recently, the phase III trial KEYNOTE-048 trial showed a higher activity with pembrolizumab plus chemotherapy versus the EXTREME regiment (based on the triplet of carbo- or cisplatin, 5-fluorouracil and cetuximab) in all patients, as well as with pembrolizumab alone in front-line of recurrent/metastatic tumours expressing PD-L1 [52].

We have previously described the importance of hypoxia in treatment response. Several studies have explored the effect of hypoxic radiosensitizer to overcome the radioresistence mechanisms induced by low oxygen levels inside the tumour mass [53,54,55,56,57]. Among them, the DAHANCA 5 trial tested the hypoxia modifier nimorazole in combination with radiotherapy on HNSCC patients [58,59]. The study results revealed that the multi-treatment was only beneficial in HPV-negative patients, with respect to HPV-positive tumours. These results suggest a possible interaction between hypoxia and HPV status in the response to radiation effects [60].

In conclusion, the different behaviour, pathological features, HNSCCs sub-groups prognosis, patients’ comorbidities, treatment side effects (especially in multidrug regimens), and the limited clinical options available make the identification of new treatment options challenging for future perspective.

## 3. Precision Medicine in HNSCCs

Precision medicine represents an emerging therapeutic approach focused on tailoring treatment for each individual patient. This discipline is based on the concept that the prediction of clinical outcomes is more accurate if driven by analysis on the features of single tumour heterogeneity, rather than by comparison between cohorts of patients [61]. This concept has been developed in two major approaches (Figure 1). The first one is based on big OMICS data analysis to predict tumour behaviours. Here, multiple omics analysis, including epigenetics, genomics, metabolomics, and proteomics, are performed on tumour samples, in order to characterise the molecular and genetic properties of every single tumour [61,62]. The second approach consists of the development of personalised preclinical platforms using in vitro and in vivo techniques to test behaviours and drug sensitivity of patient-derived samples [63,64]. In this review, we will summarize the main techniques and models available to study and improve the efficacy of precision medicine in the treatment of HNSCCs.

### 3.1. Genomic Approach

The rise of next genome sequencing (NGS) techniques has allowed for the collection of large sets of omics data in a relatively simple and easily accessible way. The storage of genomic and clinical results represents an important opportunity for translating genomic information in clinical application. In this scenario, institutions worldwide have been contributed to the sharing of big data through international projects, such as the American Association for Cancer Research (AACR)’s project, Genomics Evidence Neoplasia Information Exchange (GENIE) [65], and the American Society of Clinical Oncology (ASCO)’s CANCERLINQ [66].

The genotype-drug matching approach has been demonstrated a valid strategy in the treatment of several tumour types. An example is the treatment of non-small cell lung cancer (NSCLC) patients expressing EGFR mutations with EGFR tyrosine kinase inhibitors (TKIs) [67]. The HNSCCs mutation landscape is largely known, but a challenging future objective is the matching of the genomic profile features with specific treatments. For this reason, new biomarker-driven trials have been designed to explore the efficacy of genotype–drug matching. The EORTC 1559 study (NCT03088059), the first international biomarker-driven study including HNSCCs, will compare the activity of targeted agents (including afatinib, palbociclib, niraparib, and entrectinib) with immunotherapy [68]. NGS technology will be used to analyse copy number variations and somatic mutations of 13 tumour suppressor genes and oncogenes; PTEN and p16 expression will be evaluated through immunohistochemistry. Patients will be divided into the different treatment arms based on genomic results and a pre-defined algorithm [68]. However, biomarker-driven clinical trials still have some limitations, such as narrow gene panels and restrictive matching algorithms, which entail a low matching rate [69]. Moreover, personalised medicine trials do not consider the possibility of combined strategies [70]. Currently, clinical trials based on personalised combined treatments are needed in HNSCCs.

The evolution of sequencing technologies has improved assay sensitivity and the analysis of biological samples. These achievements raise the opportunity to analyse the presence and the features of circulating tumour DNA (ctDNA) through liquid biopsy technologies [71]. Compared to tissue samples, the collection of liquid biopsies is less invasive, and can be performed easily and more frequently. Therefore, this type of approach allows for the monitoring of the presence of ctDNA over time, making ctDNA a potential biomarker [71]. Liquid biopsy is applied in several clinical evaluations, such as the prediction of patients’ outcomes and the selection of patients for personalised therapies [72]. In this context, the INSPIRE trial (NCT02644369) explores the correlation between ctDNA and the clinical outcomes of patients treated with pembrolizumab, which also includes HNSCCs [73]. These studies allow for a better patient selection for personalised treatments, especially for HNCs, in which patients are most likely to benefit from immunotherapy.

### 3.2. Multi-Omics Approaches

HNCs are diseases closely related to the tissue microenvironment and characterised by complex and heterogeneous molecular profiles [74,75]. For this reason, a single omic analysis represents a limited approach for describing the entire genetic and molecular profile. Over the last several years, innovative combinations of high throughput omics technologies, including transcriptomic, epigenomics, metabolomics, and proteomics, have been developed [76,77].

Among them, transcriptomic analysis explores total RNA transcripts present in the sample, such as mRNA, microRNA, and long noncoding RNA transcripts (lncRNAs). The cancer expression profile is not static, but it is affected by microenvironment and treatment stimuli [76,77,78]. The utility of transcriptomics in genotype-drug matching clinical trials represents one of the hot topics of oncology research. In this field, the WINTHER trial (NCT01856296) is an open non-randomized study matching mutation and transcriptomic profiles with the selection of therapies in metastatic cancers [79]. The two processes used, genotype- and transcriptome-matched drugs, showed similar response rates, ranging between 20% and 30% [79].

A further omic approach is represented by the study of the microbiome. Humans host thousands of microorganism species in several organs, especially in the oral cavity tissues [80]. The microbiome interacts with malignancy components through the release of metabolites and molecules, producing systemic and local effects [80,81]. The microbiome-related alterations modify cancer pathogenesis, progression, and drug response, including the efficacy of immune-checkpoint inhibitors [81]. Several studies have reported different microbiome compositions in the saliva of healthy and HNSCC patients, demonstrating a reduced risk of cancerogenesis related to the presence of specific bacteria strains [82,83,84]. The role of microbiome as a tumour biomarker and in the selection of patients that will benefit from immunotherapy is currently being studied for the NCT03686202 and NCT03838601 clinical trials [85].

The heterogeneity of the tumour microenvironment makes it necessary for researchers to discriminate between the contributions of the different cell types in cancer development. An innovative option is represented by digital spatial profile (DSP) techniques [86,87]. DSP is a high-plex spatial profiling method to spatially discern regions populated by healthy cells from those populated by malignant cells starting from formalin-fixed paraffin-embedded (FFPE) samples [86,87]. The detection of different targets is performed using oligonucleotide tags conjugated to oligo-labelled primary Abs or a cocktail of RNA probes for the in situ detection of proteins or RNA transcripts [88]. Then, a mix of fluorescently labelled Abs, markers of different cell populations, is used to visualise the targets of interest directly on the slides [88]. These systems allow for a fast and extended analysis simultaneously of up to 96 proteins or 1400 mRNA in about 48 h [86,89]. In the field of precision medicine, DSP represents a promising tool to predict prognosis and treatment responses through the detection of discrepancies in mRNA and protein expression patterns. These features are particularly attractive for the study of immunotherapy efficacy, as well as to discriminate between responder and non-responder patients. The application of DSP to the evaluation of immunotherapy response has already been explored in many solid and haematological tumours, such as non-small cell lung cancer (NSCLC), melanoma, and lymphoma [90,91,92]. Kulasinghe A. et al. was the first group of researchers to apply DSP technology to HNSCCs [87]. The study aimed to correlate immune and tumour markers to the clinical outcomes of a cohort of patients treated with nivolumab and pembrolizumab. The authors used pan-cytokeratin and two immune markers, CD3 and CD8, to discriminate between malignant- and immune-infiltrate regions inside the tumour tissues. Interestingly, the immune cell markers involved in CD8 T-cells infiltration were not predictive of outcomes in patients treated with immunotherapy. Instead, a panel of immune cell protein markers, CD4, CD68, CD45, CD44, and CD66b, was related to progressive disease. Due to the low number of patients included, the project represents an explorative study that needs to be confirmed on a higher number of patients.

Imaging technologies, such as magnetic resonance imaging (MRI), positron emission tomography (PET), and computed tomography (CT), assist clinicians in making therapeutic decisions. Innovation in image processing through artificial intelligence tools have increased the information and details available which are impossible to understand with common diagnostic technologies [93]. This innovative omics approach, named radiomics, has also been employed in HNC studies on medical imaging for treatment monitoring. Indeed, radiomics is used to identify treatment-resistant sub-volumes where radiobiological events, such as hypoxia, CSC accumulation, or a high proliferative rate, can alter the treatment efficacy [94,95]. In HNSCs, radiomics has been employed to understand the radiological differences between HPV-positive and negative patients. For example, HPV-positive tumours showed well-defined margins compared to the poorly defined boundaries of their negative counterparts [93,96]. This feature is typical of malignancies that tend to invade adjacent tissues, as described for HPV-negative HNSCC tumours.

In the field of radiomics, the technology involved in the extraction of spatial information, which is used afterwards to construct prediction model, is named dosiomics. Through machine learning techniques, dosiomics extracts characteristics from 3D radiotherapy dose distribution by intensity, textural, and shape-based characteristics that allow a high complexity description of the dose distribution [97,98]. Dosiomics is increasingly used to improve the prediction of clinical outcomes and side effects, such as the locoregional recurrence in HNSCCs after intensity-modulated radiation therapy (IMRT) or local control after radiotherapy [99].

Another multi-omics approach, which combines genomics and pharmacology to study the role of the genome in drugs response, is represented by pharmacogenomics. The correlation between genetic variants and clinical outcomes helps clinicians to stratify patients in responder and non-responder groups, to prevent side effects to specific treatments [100]. Pharmacogenomics take advantage of machine learning tools to process large amount of omics data with the aim to personalize patients’ treatment based on a genotype-guided chemotherapy [93,101]. The implementation and sharing of patients’ pharmacogenomics data are encouraged by many large-scale initiatives with the aim of developing detailed drug-gene interaction databases and platforms. Two examples are represented by the USA project Clinical Pharmacogenetics Implementation Consortium (CPIC), created by the Pharmacogenomics Knowledge Base (PharmGKB) and the National Institutes of Health (NIH), and the European Ubiquitous Pharmacogenomics program (U-PGx) [102,103].

Despite the improvements in knowledge on the molecular and genetic profiles of HNSCCs tumour tissues and immune infiltrate populations, the genomic and multi-genomic-based therapeutic decisions currently have a limited role in clinical practice. Future clinical trials and additional research are needed to better understand how to use the tumour and immune cells genomic profiles in order to drive the selection of the best treatment option for each patient, as well as to develop panels of markers that help to identify patients with high or low risks of recurrence or metastasis.

### 3.3. In Vitro and In Vivo Platforms to Develop Precision Medicine in HNSCCs

The study of HNSCCs genomic profiles is just one of the possible approaches for increasing the impact of precision medicine in clinical practice. Indeed, a multitude of preclinical models have been developed to improve and validate personalised treatments. The available models include the use of both in vitro and ex vivo samples, such as immortalized cell lines and patient-derived primary cultures, and also in vivo approaches (Table 1). In this review, we will explore the combination of ex vivo and in vivo systems in the field of HNSCCs precision medicine.

#### 3.3.1. Cell Lines and Primary Cultures in HNSCCs Preclinical Research

Currently, in vitro preclinical oncology has largely used immortalized cell lines cultured on a monolayer [104]. The loss of the mechanisms involved in proliferation control guarantees the availability of a large amount of material, allowing for the possibility of developing a high number of experimental replicates and sample storage. In the literature, around 300 human HNC cell lines are listed, the majority of them derived from HPV-negative oral cavity squamous cell carcinomas [105]. Samples typically undergo extensive adaptations to survive in the in vitro controlled microenvironment. Indeed, media compositions and plastic supports are just two of the parameters that contribute to genetic alterations and the phenotypic and morphological features of patients’ cancer cells [105,106,107,108]. The heterogeneity of tumour subclones is reduced by clone selection, with the promotion of a more homogeneous cell population. Moreover, the loss of stromal and immune cell components makes it impossible to replicate in vitro, using cancer cell lines, the microenvironment features, including molecules and tumour-stroma crosstalk, cell interactions, and the three-dimensional architecture of tumour niches [104,109,110].

Thanks to ex vivo samples isolated from patients’ tissue biopsies, it is possible to overcome some of these disadvantages. Indeed, primary cultures allow for the preservation of the malignant subclones and the genetic and phenotypic cell features [104,110,111]. All of these features actively contribute to mimicking the human tumour microenvironment, as well as its contribution to drug response [112]. Thus, primary cultures represent an important tool for translational research because the preservation of tumour microenvironment makes the data of drug efficacies more realistic than those obtained from in vitro experiments. The ex vivo samples are also widely used in HNC preclinical research. Primary HNC cultures are established through different protocols. Explant cultures, chemical, enzymatic, and mechanical dissociation represent some examples of the methods used to establish primary culture [105,108,113,114]. The main challenge in the management of HNSCC primary samples is represented by the achievement of appropriate culture conditions for cell expansion. The selection of medium (such as DMEM, DMEM-F12, or RPMI-1640), serum percentages (from 0% to 20%), and growth factors, (normally fibroblast growth factor (FGF) and epidermal growth factor (EGF)) are the main parameters to consider [104,115,116,117,118,119,120]. Usually, primary cells are cultured in serum-free medium. This condition is useful for supporting the growth of stem-like tumour cells and to interfere with cancer-associated fibroblasts (CAFs) expansion [121,122,123,124]. CAFs are considered contaminant cells due to their ability to proliferate and survive at the expense of HNSCC cells, often even under serum-free medium [125,126]. To overcome this problem, cell scraping or serial trypsinization passages, based on the CAFs’ characteristic of detaching earlier than epithelial malignant cells, represent two alternatives [105,113,127]. To date, a unique protocol for an efficient establishment of HNSCC primary cultures does not exist, and further investigations are needed to overcome this barrier.

#### 3.3.2. 3D Culture Models in HNSCCs Precision Medicine

HNCs are solid tumours, a group of malignancies that take advantage of the crosstalk between cell and non-cell components [128,129]. These interactions regulate molecular processes involved in cancer progression, migration, drug sensitivity, and proliferation, taking part in the development of a protumorigenic microenvironment [104,130,131]. Primary cultures partially mimic tumour microenvironment features. Despite the preservation of the heterogeneity of original tumour cell populations, these models lack other important features of human cancer tissues, such as the three-dimensional cell distribution. Tissues architecture contribution must be considered and included by in vitro studies in precision medicine.

Several 3D scaffold free and scaffold based in vitro models have been developed to reproduce the tumour microenvironment. Regarding scaffold-based models, several strategies have been developed using synthetic or biologically derived matrix materials. In this scenario, non-polymeric materials or decellularized matrices are two options for faithfully reproducing the natural pathophysiological tumour features [132]. Collagen is the most represented component of connective tissues [133], and, consequently, collagen-based scaffolds are widely used as excellent ECM mimetic devices. Recent studies demonstrated the capability of collagen-based scaffolds to induce different drug resistance mechanisms and pathological features on the basis of cell types [134,135]. Zhang M. et al. studied the radiosensitivity of HPV-positive and negative oropharyngeal squamous cell carcinoma (OSCC) cell lines using a synthetic polyester-based scaffold [136]. Their results show a higher radiosensitivity of HPV-positive OSCC cells than the negative counterparts, supporting better clinical outcomes of this kind of patient, and thus demonstrating the translational value of the device [136]. Moreover, we have published the first study using collagen-based scaffolds harbouring both cell lines and primary cultures to study OSCCs [131]. Our findings show the ability of the 3D system to mimic some of the characteristics of OSCCs, such as aggressiveness, migration properties, and drug sensitivity, better than the common monolayer culture [131]. Taken together, these results suggest a high potential of 3D scaffold models as drug screening platforms for the selection of the most effective treatment on patients’ derived cells. 

Tumours are described as dynamic masses in continuous evolution [134]. Indeed, malignant lesions are characterised by different areas, each displaying their own phenotypic and biological features [137]. Scaffold-free cultures are excellent options for reproducing the spatial distribution of cancer cell populations. These systems usually refer to a culture developed through a self-assembled or forced cell aggregation characterised by a spherical shape [128,138]. This physical parameter allows for the development of superficial and inner zones with different nutrients and oxygen gradients, a structural characteristic of the tumour mass [138,139,140]. Oxygen tension and permeability are features that affect the chemo- and radio-cytotoxicity on tumour cells [141,142,143]. For these reasons, scaffold-free cultures are excellent and representative systems for in vitro drug screening assays. Moreover, scaffold-free cultures can be obtained from patient-derived cells, such as organoids. These spherical cultures are defined as self-organized structures derived from stem cell subclones, able to develop a 3D architecture [144]. The possibilities of obtaining healthy and cancer tissue-derived cultures represent a powerful advantage for the development of drug screening platforms which are able to predict drug response, as well as to identify compounds targeting cancer cells rather than healthy components [144,145,146,147]. In 2011, Lim Y.C. et al. successfully generated, for the first time, organoids from HNSCC patients’ tissues (named squamospheres) [148]. The 3D cultures were obtained starting from a single cell suspension derived from the digestion of primary HNC samples and cultured in serum-free media. In this preliminary study, the authors tested four chemotherapeutic agents: cisplatin, 5-fluorouracil (FU), paclitaxel, and docetaxel [148]. Differently, Driehuis E et al. measured the effect of conventional therapies already used in clinical practice, together with others tested for future implementation, on a panel of oral mucosa- and 31 HNSCC patient-derived organoids [149]. Treatments included targeted therapies (everolimus, niraparib, alpelisib, AZD4547, and vemurafenib), chemotherapeutic agents, radiotherapy, and different combinations of the above-mentioned drugs. The organoid’s ability to predict drug response needs to be confirmed in prospective clinical trials. One such validation study is represented by ONCODE-P2018-0003, in which 80 HNSCC patients’ outcomes after standard first-line treatments will be compared to organoids drug sensibility [150].

Previously described models are generated through single cell suspensions obtained after tissue digestion. This process entails the loss of the 3D tissue architecture and the selection of some cell populations [151]. It represents an important disadvantage for many research topics, such as the study of immunotherapy effects, as the spatial interactions with TME represent a crucial point of these drug mechanisms. Patient-derived explants (PDE) are another option to perform tumour cultures established by tissue fragments. In this case, samples maintain the original ECM structure and the vascular, immune, and stromal cell populations retain the histological characteristics of the original tumour [151,152]. PDEs have been widely used in preclinical research since the early 1950s [153,154]. The success of PDE platforms to predict patient responses is supported by several studies [155,156]. For example, Vescio R.A. et al. tested the reliability of a histoculture drug response assay (HDRA) as a drug screening platform for stomach and colon cancers [155]. After the first preclinical promising results, the HDRA clinical trial showed a correlation rate of 92.1% in predicting treatment efficacy in patients affected by colorectal and gastric cancer [157]. In the field of HNSCCs, Majumder B. et al. studied the efficacy of an innovative ex vivo drug screening platform, named CANScript, in which fresh tissue samples were sliced and seeded in supports coated with matrix proteins or injected as a patient-derived xenograft (PDX) [158]. The cytotoxicity data from the two models showed a strong correlation. Moreover, the CANScript technology was also able to stratify the patients treated with the combination of docetaxel, cisplatin, and 5-fluorouracil (TPF) into partial or complete responders and non-responders [158]. Another recent study compares the drug effect of targeted therapy in tumour tissue explants of PDX and directly in mice models [159]. The results showed a significant correspondence between the drug efficacy of ex vivo analysis (TEVA) and in vivo in mice. Despite the excellent results of these platforms, PDEs are not yet applied as a support in clinical practice. 

#### 3.3.3. HNSCCs PDX in Murine Model

In vivo models of HNSCCs, named patient derived xenografts (PDX), are developed through the implantation of cancer specimens or the injection of cancer cell suspensions into host animals. Nowadays, immunocompromised mice represent the most common species used in preclinical research. Over 50 years ago, engineered mice did not yet exist, and to overcome the problem of human samples rejection, engraftments were performed using murine cell lines in syngeneic models [160,161]. Now, a number of immunodeficient mouse strains are available, characterised by different engineered methods used to produce an impaired immune system. For example, the first immunocompromised mice, BALB/c Nude (nu/nu), were obtained through a mutation in Foxn1 gene that induced a low number of T-cells [162]. In this new scenario, PDX has been gradually employed in almost all the research fields associated with pathologies, including HNSCCs. 

Initially, HNSCC PDXs were performed using athymic nude mice as hosts, but these murine models showed low engraftment rates [163]. The development of strains characterised by a higher immunodeficiency, such as NOD SCID gamma (NSG) mice, have increased the engraftment efficacy to the 80% rate of success obtained by Li H. et al., with the development of 61 PDX out on a total of 76 samples tested [164,165,166]. The scientific efforts in this field provided important translational results. For example, sample engraftments within 8 weeks correlate with a worse patient outcome, and PDXs from tumours that have developed lymph node metastasis in patients displayed higher engraftment efficacy in vivo [165,166,167,168]. Moreover, the implantation of human tissue fragments allows for the conservation of the microenvironment structure, and recapitulates the cell population’s heterogeneity [150,169]. As in the primary culture model, PDXs tend to lose progressively the genetic and physiopathological original features [165,170]. In order to demonstrate the ability of PDX to preserve the human cancer characteristics and the sensitivity to treatments, many studies have compared the genetic landscape of tumour samples or public genome databases (such as The Cancer Genome Atlas) to PDX tumours [10,165]. Copy number profiles, mutation status, and gene expression levels of PDXs and HNSCC tumours resulted in high concordance across the whole genome [10,165,167,168,169,171,172]. Thus, the HNC cell lines did not provide the same results, showing different genetic status when compared to matched human cancer tissues [165]. The same concordances were showed by comparing proteomic analyses. Despite the substitution of human stromal components with murine counterparts, the majority of proteins are conserved in PDX tumours, except for the selection of proteins related to proliferative pathways [166,173].

All of these features make PDXs in immunodeficient mice a promising model to predict patient drugs response. In Table 2, the projects that have investigated the drug sensitivity of PDXs in mice are summarized (Table 2) [167,168,170,174,175]. Chemotherapeutic agents, radiation, and several tyrosine kinase inhibitors (TKI) were the principal tested therapies. We have already reported the importance of immunotherapy in the treatment of HNSCCs. The mechanism of action of these types of drugs is related to the interaction between drugs and the immune system [176,177]. The absence of a functional immunity in immune deficient mice makes this model inappropriate for evaluating the effect of immunotherapeutic drugs. To overcome this limit, several research groups developed humanised PDX models through the co-engraftment of cancer tissue fragments and haematopoietic stem cells [178,179]. Therefore, the addition of human immune cells increases the host human systemic reproduction, essential for mimicking the complexity of the tumour microenvironment. Morton J.J. et al. developed the XactMice model, in which the injection of human hematopoietic stem and progenitor cells reconstitutes the radiation-depleted bone marrow of a NOD/SCID/IL2rg (−/−) (NSG) mice [179]. The data showed how the addition of these immune cells increased the amount of immune infiltrate, induced lymphangiogenesis, and modified the tumour gene expression profile [179]. 

**Table 2 jpm-12-00854-t002:** Summary of the methods used for patient-derived xenografts of HNC cell cultures in mice.

Mouse Strain	Cancer Types	Implantation Site	PDX Tested	Treatment Tested	Reference
NOD/Hsd: Athymic Nude-Foxn1nu	Tonsil, Base of Tongue, Floor of Mouth	Subcutaneously	3	Radiation, Cisplatin, Cetuximab	[170]
NSG (NOD.Cg-Prkdcscid Il2rgtm1Wjl/SzJ)	Tongue, Alveolar ridge, Buccal	Subcutaneously	3	Flavopiridol, Belinostat, Docetaxel	[174]
NSG	Tonsil	Subcutaneously	5	Radiation	[175]
NSG	NS	Enzymatic dissociation and single cell suspension subcutaneously injected	10	Abemaciclib	[167]
Nu/Nu/(NOG, NOD/Shi-scid/IL-2Rγnull)	Tongue, oropharynx	Subcutaneously	3	Afatinib, BKM120	[168]

NS: not specified.

However, PDX models retain some important limits. For example, the implant of a single tissue fragment does not represent the tumour cell heterogeneity [165,180]. Moreover, the implantation is performed subcutaneously in the mice’s flanks, and not orthotopically, with a consequent decrease of the translational powers of obtained results. On the other hand, orthotopic implantation into oral tissues entails many disadvantages, such as the development of a tumour mass in the oral cavity that prevents the animal from feeding [181].

In conclusion, the evidence of drug sensitivity correlation between patients and the relative derived-PDX mice shown by these studies and in larger patient cohorts demonstrates the reliability of this preclinical model as a drug screening platform for HNC precision medicine.

#### 3.3.4. HNSCCs PDX in Zebrafish Models

Preclinical drug screening platforms represent promising tools to optimize the single-patient treatment. However, these approaches are characterised by some limits, as previously discussed, that harper the translation of results in clinical practice. Certainly, the complexity to culture ex vivo samples and the low engraftment rates obtained with PDX murine models are two important experimental limits [182]. Moreover, the low amount of biological material available from surgical specimens and the temporal gap between tissue manipulation and experiment readouts contribute to making preclinical platforms difficult to apply in clinics [104].

A valid and innovative scientific approach to overcome part of these limits is represented by the zebrafish model. The first time that the zebrafish was used as host for xenografts dates back to 2006, when human melanoma and multiple myeloma cells were injected in zebrafish embryos [183,184]. Different from murine models, the zebrafish is mostly used in preclinical research at embryonic and larval stages until reaching 5 day post-fertilization (dpf). Indeed, the zebrafish is a highly fecund animal, able to lay hundreds of eggs weekly. Moreover, embryos are transparent for at least 2 days, or more if treated with PFU, and this characteristic allows for the tracing of cell dynamics very easily during embryonic and larval development. The zebrafish is also an easy-to-handle animal model, and its easy genetic manipulation is useful to develop transgenic or mutant strains [185]. Among them, the casper zebrafish is a double mutated (roy−/− and nacre−/−) strain characterised by the absence of pigmentation for the entire life [186]. This feature allows for in vivo imaging of inner organs, and makes casper zebrafish an optimal recipient to trace cell dynamics. Zebrafish manipulation is simpler compared to mice, and gives the opportunity to inject hundreds of embryos per day by a single operator [182,187]. Another advantage is represented by the low number of cells necessary for each injection (normally between 100 and 500 cells per embryo), or the small tissue fragments necessary for each single transplantation [188]. For all of these reasons, the zebrafish model is particularly suitable for large scale and high throughput PDX studies.

The use of PDXs in zebrafish is a recent approach, and the first documented reports of drugs tested on embryos injected with human primary cells are not older than a decade [184,189]. Since then, the use of xenotransplantation of primary cultures in zebrafish larvae has been documented by many studies. The development of innate immunity starts in the early embryonic stages, and ends in around 2/3 weeks [190]. Therefore, if mice need to be immunocompromised to avoid human cells rejection, zebrafish embryos have an incomplete immune system that allows xenotransplantation at embryonic stages. Recently, several groups have tested PDX in zebrafish for many solid and haematological malignancies [191,192,193,194,195,196,197,198,199,200,201,202,203,204,205,206,207]. The first documented PDX of HNSCCs cells was performed by the Al-Samadi A. et al. group in a preliminary study based on the development of a PCR-based assay to measure drug efficacy [207]. The project aim was the achievement of a fast method to screen a high number of compounds based on the quantification of human DNA inside fish as an indirect count of viable cells after drug exposure. Despite the low number of samples tested (four HNC cell lines and a metastatic oral tongue cancer primary culture), it represents the first and promising attempt to develop a drug screening platform for HNC personalised medicine [207].

The more common samples xenotransplanted in zebrafish are represented by commercial cell lines and single-cell suspensions obtained by the digestion of patient tissues [152,182]. However, in contrast to the murine model, primary tissues can be successfully engrafted in zebrafish embryos. Marques I.J. et al. developed an easy and fast method to transplant small tissue fragments into the yolk sack of 2 dpf embryos [195]. Pancreas, colon, and stomach cancer tissues were monitored in the embryo’s body to test the metastatic potential of the primary cells [195]. Usai A. et al., for the first time, studied the feasibility of a drug screening platform using zebrafish embryos transplanted with malignant patient tissues, named Avatars, and they developed a method to identify the dose conversion criterion from human to zebrafish [208]. Avatars were created using tissues from 24 patients affected by pancreatic, colon, and gastric cancers, and different chemotherapeutic regimens were tested to measure the fragments area regression [208]. The samples belonged to patients enrolled in NCT03668418 (xenoZ) co-clinical trials, an observational prospective study based on the xenotransplantation of primary hepato–biliary–pancreatic and gastro–intestinal cancer samples in zebrafish embryos [209]. The XenoZ model demonstrated the prediction of the patient’s clinical outcomes, discriminating between responders and non-responders patients. Here, it can be considered a pioneering study focused on the reliability of co-clinical trials based on the use of zebrafish for the selection of personalised therapies. Transplantations of HNCs tissue fragments have not yet been reported in scientific literature, and they also represent an important limitation for the development of precision medicine platforms in these aggressive forms of cancers.

## 4. Conclusions

In this review, we have summarized the current approaches and the promising technologies in the field of personalised medicine for HNC patients. Genomic and drug screening platforms represent the most used options which are currently achievable, but new technologies and preclinical models are under development, which will be explored worldwide. It will be important to overcome the disadvantages of the current system, starting from the progress of innovative approaches which are able to study immunotherapeutic agents. However, we think that collaborations and data sharing represent the best scientific strategies for obtaining successful improvements of the current and future tools in precision medicine.

## Figures and Tables

**Figure 1 jpm-12-00854-f001:**
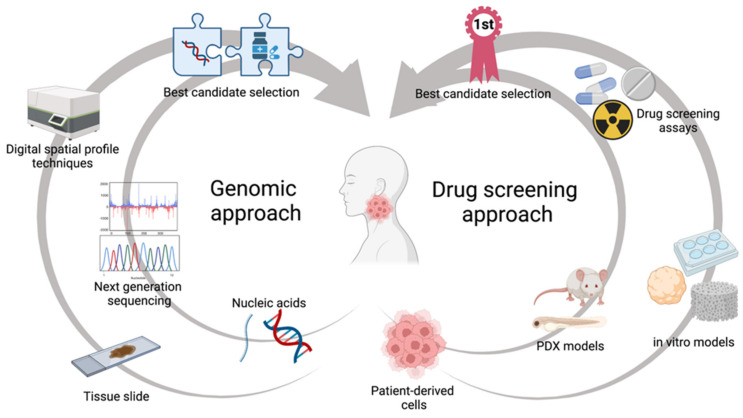
The two main precision medicine options in HNC oncology: genomic and drug screening approaches. Image created with BioRender.com.

**Table 1 jpm-12-00854-t001:** Main characteristics of in vitro and in vivo preclinical cancer models.

	2D Cell Line Cultures	2D Primary Cultures	Biomimetic Scaffold Cultures	Organoids	Mouse PDXs	Zebrafish PDXs
Easy manipulation	+++	++	++	+	+	+
Cost	Very low	Very low	Low	Medium	High	Low
Experiment duration	Days	Days	Days	Days/Months	Months	Days
Number of cells needed for drug screening assays	Low	Low	High	Medium	High	Low
Drug screening throughtput	+++	+++	++	+++	+	+++
Cancer subclones conservation	/	+	+	+	++	++
TME concervation	/	+	++	++	++	++
Immune components conservation	/	+	+	+	+	+

(+) sufficient, (++) good, (+++) optimal, (/) not suitable.

## Data Availability

Not applicable.

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
