# Peer review of "Precision Medicine in Head and Neck Cancers: Genomic and Preclinical Approaches"

_jpm, 2022, doi:10.3390/jpm12060854_

Round 1

Reviewer 1 Report

This is a topical paper focusing on HNC-omics towards treatment personalisation. The main focus of the paper is on in vitro and preclinical in vivo models developed to pursue this aim. The article is generally well-written, though it would benefit from a revision to correct the linguistic / grammatical errors that appear in the text. Below are my comments to improve the scientific complexity of the topic:

When presenting the role of cancer stem cells in HNC carcinogenesis and response to therapy, the authors should discuss the link between CSCs in HNC and HPV status, showing that genetic differences among CSCs and not their fraction (quantity) dictates the differential treatment response between HPV+ and HPV- tumours: https://pubmed.ncbi.nlm.nih.gov/27026319/

https://pubmed.ncbi.nlm.nih.gov/31015153/

A recommendation that the authors should tackle within the multi-omics section (section 3.2):

For a personalised approach to therapy, beside genomics, there are ‘omics’ elements (some better established than others) which show promising results in head and neck cancer management. While the authors briefly mention proteomics, metabolomics, and transcriptomics, there are image-based omics (radiomics), radiotherapy planning-based omics (dosiomics) and drug-based omics (pharmacogenomics) - three omics subfields that are currently under scrutiny both in pre-clinical and clinical research. A recent review published in the Journal of Personalized Medicine illustrates their potential and current applications in HNC: https://pubmed.ncbi.nlm.nih.gov/34834445/

Mention also the drug-gene interaction databases / platforms to ease the implementation of personalised chemotherapy: https://pubmed.ncbi.nlm.nih.gov/31562822/

In section 3.3.2. the authors mention the role of oxygen in tumour radiosensitisation. The hypoxic aspect of HNCs should be presented earlier in the manuscript, even in the Introduction where proliferation (EGFR) is also discussed shortly. Given the high hypoxic content of HNCs, and the known resistance to therapy due to hypoxia, biomarkers for hypoxia in view of treatment personalisation are an important facet of HNC research.

Please reword the first sentence of Conclusions (‘improvement of the scientific community interest’ does not sound right). Also replace ‘improvement’ with ‘advances’ or ’progress’. Remove or reword / correct the sentence “Unfortunately, clinical application of the systems here described is still far to be considered”.

The references are double numbered in the reference list. Please correct.

Author Response

When presenting the role of cancer stem cells in HNC carcinogenesis and response to therapy, the authors should discuss the link between CSCs in HNC and HPV status, showing that genetic differences among CSCs and not their fraction (quantity) dictates the differential treatment response between HPV+ and HPV- tumours: https://pubmed.ncbi.nlm.nih.gov/27026319/ https://pubmed.ncbi.nlm.nih.gov/31015153/

Reply: In response to the reviewer’s questions, we added a description of the genetic characteristics of HNSCC CSCs (pages 2-3). In particular, we focused on the differences and the influence of the HPV positive and negative CSCs in term of response to chemo and radiotherapy. Some pertinent references have been added (num.22-31).

A recommendation that the authors should tackle within the multi-omics section (section 3.2):

For a personalised approach to therapy, beside genomics, there are ‘omics’ elements (some better established than others) which show promising results in head and neck cancer management. While the authors briefly mention proteomics, metabolomics, and transcriptomics, there are image-based omics (radiomics), radiotherapy planning-based omics (dosiomics) and drug-based omics (pharmacogenomics) - three omics subfields that are currently under scrutiny both in pre-clinical and clinical research. A recent review published in the Journal of Personalized Medicine illustrates their potential and current applications in HNC: https://pubmed.ncbi.nlm.nih.gov/34834445/

Reply: As requested, in the paragraph “Multi-omics approaches” we described the three omics subfields reported by the reviewer (page 7). We discussed the potentialities and application in HNC patient management of radiomics, dosiomics and pharmacogenomics technologies. We briefly introduced each omic approach and reported the main contributions in the field of HNC personalized medicine. Some pertinent references have been added (num. 93-103).

Mention also the drug-gene interaction databases / platforms to ease the implementation of personalised chemotherapy: https://pubmed.ncbi.nlm.nih.gov/31562822/

Reply:  As requested, we reported the principal drug-gene interaction platforms (page 7). We mentioned the USA project Clinical Pharmacogenetics Implementation Consortium (CPIC) created by the Pharmacogenomics Knowledge Base (PharmGKB) and the National Institutes of Health (NIH) and the European Ubiquitous Pharmacogenomics program (U-PGx). We reported these projects at the end of the pharmacogenomics section in order to complete the description of this innovative omics technologies. Some pertinent references have been added (num. 102,103).

In section 3.3.2. the authors mention the role of oxygen in tumour radiosensitisation. The hypoxic aspect of HNCs should be presented earlier in the manuscript, even in the Introduction where proliferation (EGFR) is also discussed shortly. Given the high hypoxic content of HNCs, and the known resistance to therapy due to hypoxia, biomarkers for hypoxia in view of treatment personalisation are an important facet of HNC research.

Reply: As suggested, we extended the theme of the hypoxia influence in HNC diseases. We organized the paragraph starting with the description of the role of EGFR in HNC malignant processes, including angiogenesis (page 3). Then, we discussed the role of hypoxia in treatments response both in the “introduction” (page 3) and “HNCs clinical approaches” (page 4) paragraphs. We divided this topic in two different sections because, as suggested by the reviewer, the high number of clinical trials that test radiosensitizers makes necessary a mention in the HNC clinical options paragraph. Some pertinent references have been added (num. 32-41 and 53-60).

Please reword the first sentence of Conclusions (‘improvement of the scientific community interest’ does not sound right). Also replace ‘improvement’ with ‘advances’ or ’progress’. Remove or reword / correct the sentence “Unfortunately, clinical application of the systems here described is still far to be considered”.

Reply: As suggested, we have reworded the mentioned sentences in the “Conclusions” section (page 14).

The references are double numbered in the reference list. Please correct.

Reply: Thanks for the recommendation. We have corrected the reference numbering.

I would like to thank the reviewer for the thoughtful suggestions and insights, which have enriched the manuscript and produced a better, more balanced account of the research.

Reviewer 2 Report

Miserocchi et al have written a high quality and clear review of the genomic techniques and pre-clinical models used in Precision Medicine in Head and Neck Cancers (HNCs). The review starts with a description of the state of the art on anti-tumor therapies of HNCs with rich bibliography of the associated clinical trials. The authors continue with a description of the advances made by genomic techniques (and other omics) for the personalization of drug therapies for HNCs, with an opening towards new high-potential areas such as circulating tumor DNAs, the microbiota and the digital spatial tumor profiling. The review then focuses on the detailed description of the advantages and disadvantages of pre-clinical biological models, from monolayer immortalized cell cultures, to PDX mice and zebrafish, including 3D cell cultures, and their applications to HNCs.

In conclusion, this review will be of great interest for researchers and clinicians working on HNCs wishing to know the state of the art of genomic strategies and pre-clinical biological models used for precision medicine of these cancers.

Minor comments:

L300: 3.3.2. D culture models... → 3.3.2. 3D culture models...

Author Response

Miserocchi et al have written a high quality and clear review of the genomic techniques and pre-clinical models used in Precision Medicine in Head and Neck Cancers (HNCs). The review starts with a description of the state of the art on anti-tumor therapies of HNCs with rich bibliography of the associated clinical trials. The authors continue with a description of the advances made by genomic techniques (and other omics) for the personalization of drug therapies for HNCs, with an opening towards new high-potential areas such as circulating tumor DNAs, the microbiota and the digital spatial tumor profiling. The review then focuses on the detailed description of the advantages and disadvantages of pre-clinical biological models, from monolayer immortalized cell cultures, to PDX mice and zebrafish, including 3D cell cultures, and their applications to HNCs.

In conclusion, this review will be of great interest for researchers and clinicians working on HNCs wishing to know the state of the art of genomic strategies and pre-clinical biological models used for precision medicine of these cancers.

Minor comments:

L300: 3.3.2. D culture models... → 3.3.2. 3D culture models...

REPLY: Thanks for the recommendation. We have corrected the paragraph title. We thank the reviewer for his/her appreciation of our work.

Round 2

Reviewer 1 Report

The authors have adequately addressed all comments raised by this reviewer.